# An Exploratory Assessment of Pre-Treatment Inflammatory Profiles in Gastric Cancer Patients

**DOI:** 10.3390/diseases12040078

**Published:** 2024-04-16

**Authors:** Catalin Vladut Ionut Feier, Calin Muntean, Alaviana Monique Faur, Razvan Constantin Vonica, Andiana Roxana Blidari, Marius-Sorin Murariu, Sorin Olariu

**Affiliations:** 1First Discipline of Surgery, Department X-Surgery, “Victor Babes” University of Medicine and Pharmacy, 2 Eftimie Murgu Sq., 300041 Timisoara, Romania; catalin.feier@umft.ro (C.V.I.F.); murariu.marius@umft.ro (M.-S.M.); olariu.sorin@umft.ro (S.O.); 2First Surgery Clinic, “Pius Brinzeu” Clinical Emergency Hospital, 300723 Timisoara, Romania; 3Medical Informatics and Biostatistics, Department III-Functional Sciences, “Victor Babeș” University of Medicine and Pharmacy, 2 Eftimie Murgu Sq., 300041 Timisoara, Romania; 4Faculty of Medicine, “Victor Babes” University of Medicine and Pharmacy, 300041 Timisoara, Romania; alaviana.faur@student.umft.ro; 5Preclinical Department, Discipline of Physiology, Faculty of Medicine, “Lucian Blaga” University of Sibiu, 550169 Sibiu, Romania; razvanconstantin.vonica@ulbsibiu.ro; 6Oncology, Department IX-Surgery, “Victor Babes” University of Medicine and Pharmacy, 2 Eftimie Murgu Sq., 300041 Timisoara, Romania; andiana.blidari@umft.ro

**Keywords:** gastric cancer, Systemic Immune-Inflammation Index, inflammation status, prognosis, postoperative complications

## Abstract

Gastric cancer ranks as the fifth most common cancer, and the assessment of inflammatory biomarkers in these patients holds significant promise in predicting prognosis. Therefore, data from patients undergoing surgical intervention for gastric cancer over a 7-year period were analyzed. This study was retrospective and involved a preoperative investigation of six inflammatory parameters derived from complete blood counts. Statistical analysis revealed a significant increase in the leucocyte-to-monocyte ratio (LMR) (*p* = 0.048), along with a significant decrease in the number of lymphocytes and monocytes compared to patients with successful discharge. Taking into consideration patients undergoing emergency surgery, a significant increase in the LMR (*p* = 0.009), neutrophil-to-lymphocyte ratio (NLR) (*p* = 0.004), Aggregate Index of Systemic Inflammation (AISI) (*p* = 0.01), and Systemic Immune-Inflammation Index (SII) (*p* = 0.028) was observed. Regarding relapse, these patients exhibited significant increases in AISI (*p* = 0.032) and SII (*p* = 0.047). Inflammatory biomarkers represent a valuable tool in evaluating and predicting the prognosis of patients with gastric cancer.

## 1. Introduction

Gastric cancer (GC) ranks as the fifth most common cancer worldwide according to Globocan 2020, highly impacting the Asian population, which has dominated, accounting for more than two-thirds of its incidence and mortality throughout 2020, seconded by Europe [1]. Although the incidence of GC ranks as the ninth position of all malignancies in Europe, Romania seems to align with the global statistics by having GC as the sixth most common cancer in the general population and the fifth in males, with 3823 new cases and 3186 deaths registered in 2020 [2].

An invaluable tool that highly affects the early diagnostic, prognostic, and treatment of GC is active screening. Additionally, surgery plays a key role in the cure of this disease [3].

The literature emphasizes the consistent correlation between gastric cancer (GC) and alarm symptoms, indicating an unfavorable postoperative prognosis. Dyspeptic symptoms, including abdominal pain, retrosternal burning sensation, and acid reflux, are prevalent both in general medical practice and among the general population. These symptoms may signify the presence of conditions such as gastroesophageal reflux disease, peptic ulcer disease, and functional dyspepsia. Hence, a thorough assessment of these symptoms and other risk factors is necessary to determine the likelihood of a gastric cancer diagnosis and its impact on the postoperative prognosis of patients [4,5]. However, currently, there are no well-established parameters with predictive value to assess the postoperative prognostics of these patients.

Inflammatory biomarkers hold increasing promise in evaluating cancer progression. Among these markers, ratios such as the neutrophil-to-lymphocyte ratio (NLR), platelet-to-lymphocyte ratio (PLR), and monocyte-to-lymphocyte ratio (MLR) have demonstrated reliable prognostic value in GC patients. The NLR and PLR serve not only as diagnostic markers for early-stage GC, with superior sensitivity compared to traditional markers like CEA and CA 19-9 [6,7], but also as independent prognostic factors for overall survival in individuals with gastric malignancies [8]. Elevated NLR and PLR levels have been associated with poorer survival outcomes in GC patients [9].

Furthermore, inflammatory indices such as the Systemic Immune-Inflammation Index (SII) and the Systemic Inflammation Response Index (SIRI) warrant attention due to their potential to predict long-term survival [10,11]. While individual studies underscore the predictive capacity of these ratios for postoperative outcomes in GC patients, some suggest that exploring associations between these ratios or between ratios and other parameters may enhance their prognostic utility [9,12].

The present study aims to evaluate the preoperative inflammatory status in GC patients, elucidate the complex interplay between inflammation and GC, and underscore the significance of preoperative inflammation in postoperative prognostication, leveraging the aforementioned biomarker ratios.

## 2. Materials and Methods

This study was conducted according to the Declaration of Helsinki. Data collection was performed after the study received approval from the Ethical Commission of “Pius Brinzeu” Clinical Emergency Hospital (No. 444/04 March 2024).

For this retrospective investigation, we scrutinized medical records originating from patients treated at a preeminent tertiary university hospital situated in Western Romania. Specifically, our analysis focused on data sourced from individuals who underwent surgical intervention for gastric cancer at the “Pius Brinzeu” Clinical Emergency Hospital.

This study encompasses a comprehensive timeframe spanning seven years, commencing on 1 January 2016 and culminating on 31 December 2022. Inclusion and exclusion criteria were established for the study. Patients with a history of prior SARS CoV-2 infection or who developed infection during hospitalization were excluded due to the strong inflammatory response associated with the virus [13,14,15]. Additionally, as neoadjuvant chemotherapy and radiotherapy significantly impact the inflammatory system, patients who underwent these treatments were excluded based on studies indicating, for example, that a lower Systemic Immune-Inflammation Index (SII) is a positive prognostic factor in this context [16,17].

Patients included in this study presented with a morphopathological diagnosis of gastric cancer (CG) with primary tumor localization ranging from the gastroesophageal junction to the pylorus. After meeting the inclusion criteria, data were collected for comprehensive statistical analysis and interpretation.

Demographic data (gender, age, and urban/rural residence) were collected.

The preoperative blood count parameters taken into consideration were as follows:Lymphocyte (Lym);Monocyte (Mon);Neutrophil (Neu);Platelet (Pla).

Various inflammatory ratios were calculated, including the following:NLR (neutrophil/lymphocyte ratio) = Neu/Lym;LMR (lymphocyte/monocyte ratio) = Lym/Mon;PLR (platelet/lymphocyte ratio) = Pla/Lym;AISI (Aggregate Index of Systemic Inflammation) = (Neu × Mon × Pla)/Lym;SIRI (Systemic Inflammation Response Index) = (Mon × Pla)/Lym;SII (Systemic Immune-Inflammation Index) = (Neu × Pla)/Lym.

To gain insight into comorbidities, the Charlson Comorbidity Index was utilized. Surgical interventions were categorized as subtotal gastrectomy, total gastrectomy, or other procedures (including complex, extensive, atypical, and palliative procedures). Postoperative outcomes, including the presence of anastomotic fistula, were analyzed, as well as whether the intervention was palliative or curative. Relapse was also considered.

Histopathological parameters such as tumor invasion (T), lymph node invasion (N), and presence of metastases (M) were analyzed, along with the disease stage. The total duration of hospitalization, postoperative hospitalization, and discharge status were also assessed.

### Data Analysis

For the statistical analysis of this study, IBM SPSS Statistics 25 software for Windows (IBM, Armonk, NY, USA) was employed. Descriptive statistics, including measures of central tendency and dispersion, were calculated for numerical variables. Frequency tables and percentages were generated for categorical variables.

To compare two independent samples, the Mann–Whitney test was utilized, while for comparisons involving more than two samples, the ANOVA test was applied. The chi-square test was employed to highlight the differences in proportions for categorical variables. Statistical significance was defined as *p* < 0.05 for all the applied statistical tests.

## 3. Results

The study cohort consisted of 360 patients with ages ranging from 23 to 91 years and predominantly male patients. As expected, due to restricted access to clinical care, the number of cases was lower during the COVID-19 pandemic (59 in 2016, 49 in 2017, 57 in 2018, 56 in 2019, 58 in 2020, 35 in 2021, and 46 in 2022).

### 3.1. Patients Who Died vs. Patients Who Survived

The patients who died postoperatively were predominantly male, had a Charlson Comorbidity Index > 3, underwent emergency surgery, and presented advanced stages of the disease. Furthermore, these patients exhibited postoperative complications at a significantly higher rate compared to those who survived, underwent total gastrectomy at a higher proportion, and had a more advanced stage disease (Table 1).

Patients who died postoperatively presented a notable decrease in monocyte counts juxtaposed with a significant elevation in neutrophil counts. Furthermore, they manifested a marked increase in the LMR, accompanied by a significant decrease in the SIRI (Table 2).

### 3.2. Emergency vs. Elective Surgery

Patients who had an unplanned surgery were similar to those who had a planned surgery in terms of age (*p* = 0.682), sex (*p* = 0.371), living in rural areas (*p* = 0.896), and hospital stay (*p* = 0.482). However, a higher proportion of these patients presented relapse (*p* = 0.04) and post-surgery complications (*p* = 0.05), as well as a higher Charlson Comorbidity Index (*p* = 0.0012). A statistically significant smaller percentage of patients with unplanned surgery had a curative intervention (90 (54.9%) vs. 151 (77.04%), *p*-value < 0.0001), with a lower percentage of patients in Stage I (14 (8.5%) vs. 22 (11.2%)) and a higher percentage of patients in Stage IV (69 (42.1%) vs. 45 (23%), *p* = 0.0105). Patients with emergency surgery showed a different inflammation profile compared to those with planned surgery (Table 3).

Table 4 presents the variation in these parameters comparing patients that were in Stage I–III to patients in Stage IV.

### 3.3. Relapse and Postoperative Complications

The age of patients with relapse did not differ significantly (*p* = 0.653), but they exhibited a significantly higher Charlson Comorbidity Index (5.98 vs. 4.87, *p* = 0.002), compared to those without relapse. However, these patients showed a significant decrease in the number of lymphocytes (1529 vs. 1818, *p* = 0.049) and neutrophils (4801 vs. 6276, *p* = 0.013), as well as in the AISI (679.21 vs. 1098.17, *p* = 0.032) and SII (1023.69 vs. 1440.16, *p* = 0.047). There was a statistically nonsignificant increase observed in the platelet count (290,872 vs. 289,882, *p* = 0.952) and PLR (230.11 vs. 211.18, *p* = 0.456). Conversely, nonsignificant decreases were noted in the monocyte count (446 vs. 511, *p* = 0.113), NLR (3.63 vs. 4.67, *p* = 0.070), LMR (4.91 vs. 5.57, *p* = 0.326), and SIRI (91.63 vs. 104.30, *p* = 0.282). A significantly higher proportion of patients with relapse underwent palliative surgery (35 (63.5%) vs. 21 (37.5%), *p* < 0.0001) and presented with a more advanced lymph node invasion stage (36 (66.7%) vs. 155 (51.7%), *p* = 0.041), with a significantly higher proportion of patients in Stage IV (29 (52.7%) vs. 85 (28.3%), *p* = 0.002).

Among patients who developed postoperative intestinal fistula as a complication, a significantly higher proportion underwent emergency surgery (23 (60.5%) vs. 15 (39.5%), *p* = 0.05). Age did not significantly differ (*p* = 0.399), yet these patients experienced a longer total hospital stay (*p* < 0.0001) and increased postoperative hospitalization duration (*p* < 0.0001). Moreover, these patients underwent total gastrectomy at a significantly higher proportion (11 (28.9%) vs. 46 (14.3%) *p* = 0.014) and had a significantly higher mortality rate (18 (47.4%) vs. 35 (10.9%), *p* < 0.001). Notably, these patients exhibited statistically nonsignificant decreases in the lymphocyte count (1727 vs. 1779, *p* = 0.758), monocyte count (478 vs. 503, *p* = 0.667), and neutrophil count (5607 vs. 6130, *p* = 0.702), as well as the NLR (3.64 vs. 4.66, *p* = 0.09), AISI (946.61 vs. 1052.78, *p* = 0.724), SIRI (92.04 vs. 103.58, *p* = 0.443), and SII (1236.47 vs. 1400.21, *p* = 0.5252). Additionally, a nonsignificant increase in the platelet count (324,105 vs. 286,000, *p* = 0.095), LMR (5.66 vs. 5.44, *p* = 0.799), and PLR (218.33 vs. 213.58, *p* = 0.830) was observed.

## 4. Discussion

This retrospective study was conducted at the largest tertiary university hospital in Timisoara, Western Romania. It is noteworthy that gastric cancer incidence is a global concern. Focusing on Europe, studies indicate that Eastern European countries, including Romania, exhibit higher incidence and mortality rates compared to their Western European counterparts. For instance, during 2015–2019, the highest incidence rates among males were reported in Eastern Europe, the Russian Federation, and Belarus, exceeding 15 cases per 100,000 individuals, followed by Ukraine, Romania, and Bulgaria, with rates surpassing 10 cases per 100,000. Conversely, the lowest rates were observed in the United States, Sweden, Australia, and Canada, with rates of around 2–3 cases per 100,000 males [18].

The assessed inflammation ratios exhibited diverse patterns associated with specific outcomes. Concerning deceased patients, higher variations were observed in the neutrophil count, along with an elevated NLR. Additionally, decreased values were noted for lymphocyte and monocyte counts, as well as a lower SIRI. Emergency surgery patients presented with a lower monocyte count, along with an elevated neutrophil count and significant increases in the NLR, LMR, AISI, and SII. Patients with relapse exhibited a significant decrease in lymphocyte and neutrophil counts, as well as AISI and SII.

The study period also encompassed the COVID-19 pandemic, during which a decrease in the volume of surgical procedures for GC was observed. This decrease ranged from 17% to 63% in hospitals across India, with a noteworthy decrease of 50% reported in a hospital in Tokyo [19,20,21]. Moreover, European countries such as Italy reported a decline of 30% compared to the pre-pandemic era [22]. This decline was attributed to the postponement of surgical interventions, leading to patients presenting with more advanced disease stages and heightened symptomatology [23].

The inflammatory status of patients assessed through the analysis of these ratios plays a significant role in evaluating the prognosis of patients undergoing surgery for malignant pathologies. Significant changes in the potential predictive value of these preoperative ratios have been highlighted not only in patients undergoing surgery for gastric cancer but also in patients undergoing surgery for colorectal cancer [24,25,26], and even in patients undergoing lung cancer treatment [27,28]. While experimentally verified reference values for the mentioned ratios are not available, the specialized literature presents retrospective studies (with their well-known limitations) demonstrating the potential of these ratios in determining patient prognosis.

In this study, the mean age of patients undergoing surgical intervention was 65 years, with the majority being male, consistent with the existing literature. Machlowska et al. reported that men are two to three times more susceptible to this disease [29]. Androgen receptor (AR) expression is pivotal in gastric cancer initiation and advancement. AR activation by androgens initiates complex pathways stimulating cell proliferation and angiogenesis, increasing men’s susceptibility to gastric cancer. The intricate interplay of the AR with oncogenic pathways profoundly influences gastric cancer development in men. Understanding these mechanisms is crucial for targeted therapies to alleviate the burden of gastric cancer in male populations [30,31]. Deceased patients exhibited a higher Charlson Comorbidity Index, predominantly underwent palliative interventions, and underwent operation as an emergency to a significantly greater extent. The type of surgical intervention performed decisively contributed to the postoperative prognosis. Thus, those who died underwent total gastrectomy three times more often and presented intestinal fistula as a postoperative complication at an overwhelming proportion (24% vs. 6.5%). Regarding disease stage, deceased patients were significantly more likely to present in Stage IV (47.2% vs. 29%) with highly advanced T (T4 84% vs. 60.9%) and N (N3 66% vs. 50.8%) stages. Metastasis was detected in nearly half of the deceased patients (45.3% vs. 29.3% in survivors, *p* = 0.015). It is well established that total gastrectomy, along with the presence of postoperative complications, represents negative prognostic factors, significantly influencing patient survival, as highlighted by Ebihara et al., who noted that patients with postoperative complications may experience up to a 12.4% decrease in the 5-year survival rate compared to those with ideal postoperative outcomes [32]. Associated pathologies, acid–base and electrolyte imbalances, and anemia, as well as more advanced T or N stages, are considered poorer prognostic factors as well [31,33].

The immune-inflammatory response has been implicated in enhancing angiogenesis; promoting tumor cell proliferation, invasion, and metastasis; and inhibiting immunomodulatory responses, all closely linked to tumor occurrence and progression [6,34]. This study highlights significant differences in this regard. Specifically, the number of lymphocytes and monocytes was significantly lower in deceased patients (Table 2). Therefore, we can argue that hematologic balance, along with patients’ response to surgical treatment, plays an important role in their prognosis. Zhang et al. specified that lymphocyte function is pivotal in inhibiting tumor cell proliferation and migration rates, thus contributing significantly to tumor immunosurveillance. Decreased lymphocyte levels directly correlate with a suppressed immune response, exacerbating the risk of GC occurrence [6]. Additionally, deceased patients exhibited a significant increase in the number of neutrophils (*p* = 0.034), a finding consistent with specialized studies showing that patients with gastric cancer typically have higher neutrophil counts compared to the general population [35].

Statistical analysis revealed an increase in the NLR and PLR among postoperative deceased patients, although the increase was nonsignificant (*p* = 0.082, *p* = 0.9780). However, the specialized literature and previous studies mention the predictive quality of these parameters and their association with a reserved prognosis [8]. Kim et al. specified that an elevated NLR (>2.5) and a high PLR (>158) are correlated with adverse outcomes in GC. The elevated NLR may reflect an inflammatory response dependent on increased neutrophil levels and a reduced immune response mediated by lymphocytes against tumors, leading to tumor progression and an unfavorable prognosis [36]. Additionally, an elevated PLR is another important indicator of systemic inflammation. The increase in platelet count may be a response to inflammatory mediators secreted by tumors or circulating inflammatory cells and is believed to contribute to the regulation of tumor angiogenesis, leading to disease progression [37].

However, it is worth noting the significant increase in the LMR (*p* = 0.048) and the decrease in the SII among deceased patients (*p* = 0.001). Regarding the LMR in this context, a mean value of 4.53 was observed among patients, while deceased patients exhibited a significantly increased LMR of 7.34 (*p* = 0.034). Analyzing this aspect reveals how an increase in this ratio corresponds to a more unfavorable prognosis for the patient. The specialized literature supports this idea, with studies revealing the LMR’s predictive value for surgical outcomes. Pan et al. observed that patients with a low LMR (≤3.37) had higher hazard ratios for overall survival (HR: 2.10; *p* < 0.001) and disease-free survival (HR: 2.46; *p* < 0.001) [9]. Additionally, Hsu et al. found that patients with a lower LMR (≤4.8) exhibited more aggressive tumor behavior, higher surgical mortality rates, and worse long-term survival (HR: 1.36, 95% CI: 1.08–1.69) [38].

It is noteworthy that patients who experienced relapse exhibited a significant increase in the SII (*p* = 0.047), as did patients who underwent emergency surgery (*p* = 0.028). These findings are supported by the specialized literature, which indicates an intricate interplay between cancer and inflammation, highlighting that high SII levels correlate with worse survival [17]. In the tumor microenvironment, inflammatory cells significantly influence carcinogenesis and angiogenesis. Neutrophils impede immune function, aiding tumor progression and metastasis via cytokine secretion and immune suppression, while platelets protect circulating tumor cells, promoting their survival and metastasis, as well as angiogenesis [16]. Lymphocytes play a crucial role in tumor control by secreting cytokines and inhibiting tumor cell proliferation and migration. Therefore, the SII emerges as a superior objective indicator of host inflammatory and immune status compared to other prognostic indexes such as the NLR, MLR, and PLR [16,17,39].

A significant decrease in the SIRI (*p* = 0.001) was observed in the case of deceased patients. This aspect contradicts the studies in the literature, which show that an increase in this ratio leads to a poorer prognosis. With the aforementioned reservation and the need for more extensive studies on significantly larger cohorts, it must be emphasized that in current studies, the SIRI emerges as a superior prognostic factor for overall survival among patients undergoing post-radical gastrectomy, surpassing other inflammatory indexes. Dynamic changes in the SIRI pre- and post-surgery accurately reflect tumor progression and therapy response. Moreover, the SIRI serves as an “immunologic signature” in gastric cancer, potentially predicting responses to immunotherapy and guiding clinical decisions effectively [10,12,40].

Last but not least, this study aimed to explore the AISI and its variation among deceased patients or those who experienced relapse. Thus, a significant increase was observed in the case of patients in the latter category (*p* = 0.032), as well as a significant increase in patients who underwent emergency surgery (*p* = 0.01). These patients arrived at the hospital with complications of cancerous pathologies; significant hematologic and acid–base imbalances; and, of course, a more reserved prognosis.

After presenting our study results and reviewing the relevant literature, it becomes evident that variations in inflammatory status parameters hold significant prognostic value for these patients. Pan et al. highlighted the preoperative predictive value of the LMR, which correlates with shortened long-term survival. Meanwhile, Kim et al. emphasized the predictive superiority of the PLR, linked to reduced survival and increased relapse. However, the SII emerges as the most effective predictor for long-term survival, surpassing the NLR and PLR. Its comprehensive reflection of inflammatory and immune responses positions SII as pivotal in prognostic assessments for gastric cancer, offering valuable insights [16,17]. These findings underscore the intricate relationship between inflammatory markers and outcomes in GC patients, providing crucial prognostic implications. Additionally, the correlation between the Charlson Comorbidity Index; the type of surgery performed; postoperative mortality; and elevated levels of the NLR, PLR, AISI, and SII underscores the multifaceted impact of these markers, enhancing their clinical relevance in the postoperative setting [6,8,16,17,34,36,37,39].

### Study Limitations

When considering the limitations of this study, it becomes clear that they are multifaceted. Primarily, we must address the retrospective nature of the study, which was conducted exclusively at a single tertiary university hospital in Romania, thereby restricting the generalizability of our findings. Additionally, the COVID-19 pandemic significantly influenced the frequency of surgical interventions, leading to patient postponements and subsequent presentation at more advanced disease stages with associated complications.

Moreover, it is imperative to conduct a comprehensive assessment of potential confounding factors, including those inadvertently overlooked in our analysis. Examining the dynamic fluctuations in ratios before and after interventions could yield valuable insights into their clinical significance. The significant contributions of this study in elucidating the role of inflammation in gastric cancer underscore the ongoing need for research to identify and validate relevant inflammatory markers with both prognostic and therapeutic potential. The extensive body of research elucidating the predictive capacity of these parameters prompts the medical community to establish rigorous criteria and methodologies for conducting experimental and prospective studies aimed at delineating reference values for said parameters. Such an endeavor seeks to mitigate inherent limitations and potential biases, ultimately facilitating the formulation of protocols that hold substantial potential to influence the management of these cases significantly.

## 5. Conclusions

This study presents significant findings regarding variations in the investigated ratios and the prognosis of patients. Thus, in the case of deceased patients, an increase in the NLR, LMR, AISI, and SII, as well as in the number of neutrophils, alongside a decrease in the number of lymphocytes and monocytes, could be observed. Furthermore, in patients undergoing emergency surgery, significant increases in the NLR, LMR, AISI, and SII were evident, with some of these parameters showing elevated values also in patients with relapse (AISI and SII) compared to those without. Due to the multitude of studies demonstrating the prognostic capacity of these parameters, the development of a standardized protocol for conducting prospective studies, considering multiple associated variables and their influence on the outcome, is essential. However, this study also presents and validates the significant impact that these inflammatory markers have on the clinical and therapeutic prognosis of patients with gastric cancer.

## Figures and Tables

**Table 1 diseases-12-00078-t001:** Characteristics of patients.

Characteristic	All, *n* = 360	Dead, *n* = 53	Alive, *n* = 307	*p*-Value
Age, years Mean(M) ± Standard Deviation (SD)	65.23 ± 10.63	67.49 ± 9.64	64.84 ± 10.76	0.0738
Sex, men	246 (68.3%)	39 (73.6%)	207 (67.45)	0.373
Rural	155 (43.1%)	28 (52.8%)	127 (41.4%)	0.120
Charlson > 3	232 (64.4%)	24 (68.6%)	183 (59.6%)	0.005
Emergency	164 (45.6%)	36 (67.9%)	128 (41.7%)	<0.0001
Curative surgery	209 (58.05%)	26 (49.1%)	215 (70%)	0.003
Type of surgery				0.001
Subtotal gastrectomy	210 (58.3%)	19 (35.8%)	191 (62.2%)	
Total gastrectomy	57 (15.8%)	16 (30.2%)	41 (13.4%)	
Other interventions	93 (25.8%)	18 (34%)	75 (24.4%)	
Complications	38 (10.6%)	18 (34%)	20 (6.5%)	<0.0001
Stage				0.02
I	36 (10%)	1 (1.9%)	35 (11.4%)	
II	57 (15.8%)	4 (7.5%)	53 (17.3%)	
III	148 (41.1%)	22 (41.5%)	126 (41%)	
IV	114 (31.7%)	25 (47.2%)	89 (29%)	
pT				0.008
1	22 (6.1%)	1 (1.9%)	21 (6.8%)	
2	25 (6.9%)	1 (1.9%)	24 (7.8%)	
3	76 (21.1%)	5 (9.4%)	71 (23.1%)	
4	231 (64.2%)	44 (84%)	187 (60.9%)	
pN				0.027
0	76 (21.1%)	4 (7.5%)	72 (23.5%)	
1	44 (12.2%)	4 (7.5%)	40 (13%)	
2	43 (11.9%)	8 (15.1%)	35 (11.4)	
3	191 (53.1%)	35 (66%)	156 (50.8%)	
pM	114 (31.7%)	24 (45.3%)	90 (29.3%)	0.015
Hospitalization, days	16.02 ± 9.90	16.51 ± 14.5	15.94 ± 8.9	0.77
Post-surgery, days	12.3 ± 9.25	12.79 ± 14.17	12.21 ± 8.14	0.78

**Table 2 diseases-12-00078-t002:** Evaluated biomarkers by groups presenting the mean value ± SD.

Marker	All, *n* = 360	Dead, *n* = 53	Alive, *n* = 307	*p*-Value
Lymphocytes	1774 ± 1887	1496 ± 618	1822 ± 2025	0.024
Monocytes	501 ± 344	391 ± 301	520 ± 347	0.007
Platelets	290,000 ± 131,871	267,103 ± 109,439	294,005 ± 135,140	0.115
Neutrophils	6083 ± 5669	8509 ± 12,660	5805 ± 4179	0.034
NLR	4.53 ± 5.04	6.30 ± 8.60	4.33 ± 4.45	0.082
LMR	5.46 ± 7.47	7.34 ± 13.45	5.14 ± 5.82	0.048
PLR	214.08 ± 176.78	214.71 ± 127.37	213.97 ± 184.162	0.978
AISI	1043.35 ± 2039.87	1217.54 ± 2490.99	1023 ± 1988.52	0.673
SIRI	102.36 ± 116.87	71.27 ± 62.71	107.74 ± 123.14	0.001
SII	1385.67 ± 1998.57	1564.23 ± 2243.62	1365.21 ± 1974	0.659

**Table 3 diseases-12-00078-t003:** Inflammation status of patients related to the type of surgery presenting the mean value ± SD.

Marker	Elective Surgery, *n* = 165	Emergency Surgery, *n* = 195	*p*-Value
Lymphocytes	1825 ± 2290	1731 ± 1255	0.573
Monocytes	545 ± 290	448 ± 392	0.007
Platelets	288,282 ± 127,688	292,116 ± 137,044	0.784
Neutrophils	5225 ± 3695	7774 ± 8058	0.013
NLR	3.67 ± 3.73	6.22 ± 6.65	0.004
LMR	4.50 ± 5.80	6.61 ± 8.95	0.009
PLR	201.47 ± 148.22	229.08 ± 205.11	0.180
AISI	787.35 ± 984.49	1548.24 ± 3188.86	0.01
SIRI	109.45 ± 93.85	93.93 ± 139.22	0.211
SII	1098.90 ± 1058.66	1951 ± 25	0.028

**Table 4 diseases-12-00078-t004:** Inflammation status of patients in Stage I–III compared to those in Stage IV (mean ± SD).

Marker	Stage I–III, *n* = 246	Stage IV, *n* = 114	*p*-Value
Lymphocytes	1897 ± 2246	1505 ± 678	0.013
Monocytes	525 ± 338	454 ± 358	0.08
Platelets	289,049 ± 127,390	286,310 ± 126,040	0.707
Neutrophils	5893 ± 4409	6622 ± 8112	0.511
NLR	4.21 ± 4.43	5.43 ± 6.35	0.176
LMR	5.72 ± 8.78	4.91 ± 3.51	0.217
PLR	204.52 ± 169.03	231.1 ± 187.95	0.202
AISI	943.26 ± 1571.57	1322.03 ± 2928.65	0.345
SIRI	104.07 ± 117.85	99.43 ± 117.48	0.729
SII	1256.82 ± 1593.75	1740.20 ± 2779.59	0.208

## Data Availability

The datasets used and/or analyzed during the current study are available from the corresponding author upon reasonable request.

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
