# Peer review of "An Exploratory Assessment of Pre-Treatment Inflammatory Profiles in Gastric Cancer Patients"

_diseases, 2024, doi:10.3390/diseases12040078_

Round 1
Reviewer 1 Report
Comments and Suggestions for Authors
The authors have mined seven years of data from patients who underwent surgical intervention for gastric cancer to evaluate a panel of inflammatory parameters as possible biomarkers of the likelihood of surgical success, risk of relapse or other debilitating complications, and possible survival time post-surgery. Decreased numbers of lymphocytes and monocytes in combination with an elevated leucocyte-to-monocyte ratio (LMR) pre-surgery were linked to poor outcomes. Increased LMR, neutrophil-to-lymphocyte ratio (NLR), Aggregate Index of Systemic Inflammation (AISI), and Systemic Inflammation Response Index (SII) were observed in patients requiring emergency surgery. The risk of relapse was highest in patients with elevated AISI and SII. The authors suggest that these Inflammatory biomarkers may provide a tool for evaluating the likely prognosis of patients undergoing gastric cancer surgery.
The findings are intriguing, but the number of patients reviewed is small and only from one hospital. More details about the hospital [location, capacity, and catchment area] would help better contextualise the data. Are patients exclusively from the locality or many regions of the country? If the former, are gastric cancer rates higher than in other parts of the country? If available, give some comparisons of incidence data from other countries in the region.
The data suggests that the incidence of gastric cancers in men is higher and more severe in men than in women. Do the authors consider there to be a genuine gender difference in susceptibility to cancer, or does greater reluctance in males to report or delay in reporting health issues skew the data? This information could be significant in developing suitable therapeutic strategies.
The serum parameters have been measured before surgery. What was the health history of patients before diagnosis of gastric cancer and surgery? Did gastric problems appear quite soon before diagnosis, or was there a prolonged period of gastric issues before diagnosis? If the latter, many of the observed immune cell changes may have been prevalent or established well before the need for surgery and may be suitable targets for some nutritional or drug therapy in the days prior to surgery.
The study encompassed patients with stage 1-4 cancers. Presumably, those with stage 4 cancers had the worst prognosis. Were the immune-cell parameters pre-surgery worse for these patients than those with lower-grade cancers?
How unique are the present immune cell findings to gastric cancers? Are they observed in whole or part with other disorders, i.e., how predictive are they of gastric problems?
If the present immune cell findings are predictive of poor gastric cancer outcome, can the authors suggest treatments or therapies to overcome the poor prognosis?
Ln 47-50 Clarify. Is it suggested that these parameters may be linked to a greater risk of gastric cancer and poor prognosis?
Ln 52-65 How does the present study seek to strengthen the tentative observations already in the literature?
Ln 94 Neutrophil.
Ln 154 ‘The age of patients with relapse did not differ significantly (p=0.653), but they exhibited a significantly higher Charlson Comorbidity Index (5.98 vs 4.87, p=0.002).’
Compared with what?
Author Response
The findings are intriguing, but the number of patients reviewed is small and only from one hospital. More details about the hospital [location, capacity, and catchment area] would help better contextualise the data. Are patients exclusively from the locality or many regions of the country? If the former, are gastric cancer rates higher than in other parts of the country? If available, give some comparisons of incidence data from other countries in the region.
The study was conducted at the largest tertiary University hospital in Western Romania, with a limited sample size of patients. It is pertinent to acknowledge the existence of other hospitals whose databases were not incorporated into this study, notably presenting significantly smaller numbers compared to our institution. Precise data regarding the incidence of gastric cancer pathology across different regions of the country was unavailable; however, according to the National Institute of Statistics, mortality rates attributable to malignant tumors in Timis County (227.0‰00) and neighboring counties slightly exceed the national average (223.0‰00). Nonetheless, lacking specific data delineating the situation concerning gastric cancer pathology, we opted not to include these statistics in the article. Nevertheless, we appreciate the suggestion to explore incidence rates in neighboring countries and have observed that Eastern European countries exhibit a higher incidence compared to their Western counterparts, an observation incorporated into our article.
The data suggests that the incidence of gastric cancers in men is higher and more severe in men than in women. Do the authors consider there to be a genuine gender difference in susceptibility to cancer, or does greater reluctance in males to report or delay in reporting health issues skew the data? This information could be significant in developing suitable therapeutic strategies.
Indeed, a highly significant aspect extensively debated within the literature. Although the aim of this study is not to investigate the factors contributing to a higher incidence of gastric cancer in the male population, we have included in the discussion section a few remarks regarding potential justifications for this phenomenon.
The serum parameters have been measured before surgery. What was the health history of patients before diagnosis of gastric cancer and surgery? Did gastric problems appear quite soon before diagnosis, or was there a prolonged period of gastric issues before diagnosis? If the latter, many of the observed immune cell changes may have been prevalent or established well before the need for surgery and may be suitable targets for some nutritional or drug therapy in the days prior to surgery.
Indeed, a very important remark. Unfortunately, as this is a retrospective study spanning a longer period of time, we did not have access to details regarding patients' symptoms prior to surgery or their preparation. Furthermore, the healthcare system in Romania is undergoing modernization; therefore, screening has a relatively low rate. Additionally, patients typically present to the hospital only when symptoms persist. Thus, we wholeheartedly agree with your suggestion that patients should undergo clinical and paraclinical preparation before such surgical interventions. We sincerely appreciate your input, and in conducting our next prospective study, we will consider monitoring these patients and adjusting preoperative nutritional and hydroelectrolytic balance, among other factors. This will enable us to truly assess the importance and influence of these aspects on subsequent outcomes.
The study encompassed patients with stage 1-4 cancers. Presumably, those with stage 4 cancers had the worst prognosis. Were the immune-cell parameters pre-surgery worse for these patients than those with lower-grade cancers?
We added a new table which present the situation in patients with stage I-III compared to patients with Stage IV
How unique are the present immune cell findings to gastric cancers? Are they observed in whole or part with other disorders, i.e., how predictive are they of gastric problems?
We did add to the discussion part a paragraph that describes the usage of these finding and their potential prognostic values in colorectal cancer as well, including a study made in our hospital
If the present immune cell findings are predictive of poor gastric cancer outcome, can the authors suggest treatments or therapies to overcome the poor prognosis?
The extensive body of research elucidating the predictive capacity of these parameters prompts the medical community to establish rigorous criteria and methodologies for conducting experimental and prospective studies aimed at delineating reference values for said parameters. Such an endeavor seeks to mitigate inherent limitations and potential biases, ultimately facilitating the formulation of protocols that hold substantial potential to influence the management of these cases significantly.
Ln 47-50 Clarify. Is it suggested that these parameters may be linked to a greater risk of gastric cancer and poor prognosis?
We did add an explanation
Ln 94 Neutrophil.
Corected
Ln 154 ‘The age of patients with relapse did not differ significantly (p=0.653), but they exhibited a significantly higher Charlson Comorbidity Index (5.98 vs 4.87, p=0.002). We did add an explanation and detalils.
Thank you very much for your comments
Kind Regards,
Dr. Muntean C
Reviewer 2 Report
Comments and Suggestions for Authors
This manuscript aims to use inflammatory conditions to predict the survival rate of gastric cancer patients. To reach such aim, the authors retrospectively studied 360 patients and the ratios of their immune cells. Although text has many irrelevant contents, manuscript presentation is clear and English language is good. However, there are three points that need to be taken care of in this study. First, this study lacks a control group as a reference. It is possible that other diseases also have such changes in those ratios and indexes. In this case, we cannot exclude the possibility that those ratios/indexes as biomarkers are not specific for gastric cancer. Second, the link between inflammatory ratios (or indexes) and gastric cancer is not verified experimentally. Authors might need to see if patients in other places have such inflammatory profiles. Without verification, conclusions in this study might not be applicable to gastric cancer patients at other hospitals. Third, the samples might not be representative for such study. The patients vary on their age, healthy conditions, sex, food intake, and some medical treatments. In sum, this manuscript has a clear presentation, but this study needs a more rigid experimental design in their research.
Comments on the Quality of English LanguageEnglish is fine.
Author Response
Q1.Although text has many irrelevant contents, manuscript presentation is clear and English language is good. However, there are three points that need to be taken care of in this study.
First, this study lacks a control group as a reference. It is possible that other diseases also have such changes in those ratios and indexes. In this case, we cannot exclude the possibility that those ratios/indexes as biomarkers are not specific for gastric cancer.
The study is retrospective in nature, presenting data obtained from a hospital in Timisoara, Romania. Despite the absence of a control group, it is noteworthy that the existing literature supports the results obtained within our study, showing similar outcomes in countries across different continents. It has not been specified that these markers are specific to gastric cancer, as it is well-established that the inflammatory system plays a significant role in the progression of any tumor pathology. Furthermore, we have added a paragraph in the "Discussion" section addressing this aspect.
Q2. Second, the link between inflammatory ratios (or indexes) and gastric cancer is not verified experimentally. Authors might need to see if patients in other places have such inflammatory profiles. Without verification, conclusions in this study might not be applicable to gastric cancer patients at other hospitals.
Q3. Third, the samples might not be representative for such study. The patients vary on their age, healthy conditions, sex, food intake, and some medical treatments. In sum, this manuscript has a clear presentation, but this study needs a more rigid experimental design in their research.
A1&A2. Currently, there are no globally established experimental studies to clearly establish and highlight reference values between these parameters and patient outcomes.. This study aims to present the impact of inflammatory status on postoperative outcomes and prognosis of gastric cancer patients, with important descriptive results. Our team presents the situation from a Tertiary University Hospital in Timisoara, Romania. We are aware of the limitations of this study, especially the small number of patients who met the inclusion criteria. A prospective study with a control group, along with prospective methodology, would be much more informative from a medical standpoint. However, we cannot deny the importance of the highlighted results, and the correlation with literature studies conducted in various parts of the world serves to confirm the obtained results. Certainly, we are keenly aware of these considerations and earnestly contemplating the pursuit of a future prospective study to establish benchmarks that would significantly contribute to the clinical and therapeutic management of these patients, in light of the findings obtained.
Thank you very much! We hope to have answered and solved the mentioned issues.
With kind regards,
Dr. Calin Muntean
Round 2
Reviewer 1 Report
Comments and Suggestions for Authors
All issues raised have been robustly and satisfactorily dealt with by the authors.